# Assessment of Existing Fate and Transport Models for Predicting Antibiotic Degradation and Transport in the Aquatic Environment: A Review

Temesgen Zelalem Addis [1] , Joy Tuoyo Adu [1,*] , Muthukrishnavellaisamy Kumarasamy [1,2] and Molla Demlie [3]

1    Civil Engineering Programme, School of Engineering, University of KwaZulu-Natal, Durban 4041, South Africa
2    Saveetha School of Engineering, Saveetha Institute of Medical and Technical Sciences, Chennai 600072, India
3    Department of Geological Science, University of KwaZulu-Natal, Durban 4041, South Africa
*    Correspondence: aduj@ukzn.ac.za

**Abstract:** In recent years, the use of antibiotics for human medicine, animal husbandry, agriculture, aquaculture, and product preservation has become a common practice. The use and application of antibiotics leave significant residues in different forms, with the aquatic environment becoming the critical sink for accumulating antibiotic residues. Numerous studies have been conducted to understand antibiotic removal and persistence in the aquatic environment. Nevertheless, there is still a huge knowledge gap on their complex interactions in the natural environment, their removal mechanism, and the monitoring of their fate in the environment. Water quality models are practical tools for simulating the fate and transport of pollutant mass in the aquatic environment. This paper reports an overview of the physical, chemical, and biological elimination mechanisms responsible for the degradation of antibiotics in natural surface water systems. It provides an in-depth review of commonly used quantitative fate models. An effort has been made to provide a compressive review of the modeling philosophy, mathematical nature, environmental applicability, parameter estimation, prediction efficiency, strength, and limitation of commonly used environmental antibiotic fate models. The study provides information linking paradigms of elimination kinetics and their simulation in the antibiotic fate models aiming at critical issues regarding current model development and future perspectives and to help users select appropriate models for practical water quality assessment and management.

**Keywords:** antibiotics; degradation; fate; kinetics; modeling; simulation

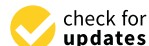



## 1. Introduction

The presence of antibiotics in the aquatic environment is regarded as an emerging contaminant and raises increasing concern worldwide due to their ubiquity and potential chronic toxicity. Antibiotic use in medicine, animal husbandry, agriculture, aquaculture, and product preservation has become common practice in human life [1]. Following their use and application, antibiotic residues enter the aquatic environment directly and indirectly in their active antimicrobial or metabolite form.

The presence of antibiotic residues in the aquatic environment, exceeding predicted no-effect concentrations and natural purification capacity, alters the physiological, reproductive, and genetic development of aquatic organisms [2]. Due to their acute toxicity, genotoxicity, pathogen resistance development, and endocrine disruption, the presence of antibiotics in water systems has raised concerns about their impact on public health and the environment [3,4]. Hence, antibiotic contamination in surface water bodies has become a primary research focus as many studies worldwide have reported their associated impact [5–10].

Emerging contaminants often occur at low concentrations and complex matrices [11]. Numerous laboratory studies have reported the inability of most analytical instruments to capture and quantify low levels of antibiotics [12–16]. Hence, continuous tracking of the level of contaminants in the environment is essential for sustainable water management. However, direct measurement of contamination levels requires considerable investment, extensive laboratory technology, and labor. Consequently, analytical and mathematical methods are employed in these efforts. Moreover, analytical determination of low concentrations of antibiotics necessitates sophisticated high-resolution equipment, which requires specialists to operate and is not easily affordable.

Mathematical modeling and simulations are effective tools for studying the fate and transport of contaminants. Mathematical models can evaluate low, barely detectable antibiotic concentrations and continuously observe (temporal coverage) contamination levels in the aquatic environment [17,18]. Additionally, the development and use of a mathematical model describing the physical, chemical, and biological interaction processes provide the ability to evaluate "what if" scenarios and to compare exposure assessments (human and microorganism). Moreover, these models provide greater spatial coverage and can be used for observations where accessibility is difficult. However, model predictions have many uncertainties regarding model input values, simplifications of complex processes, calibration, and corroboration [19]. Mathematical modeling faces challenges in representing the complex interaction between contaminants and the environment, requiring further development through continuous improvement and investigation.

A pollutant that enters a river system undergoes numerous events during transport as it interacts with many biological, environmental, and weather elements. In instantaneous or long-period interaction, the pollutant mass entering the water bodies might be either lost through various in-stream reaction mechanisms or transformed into other compounds via chemical and biochemical reactions. The pollutant mass in waterbodies can also escape by evaporation, volatilization, sedimentation, and flow abstractions without changing its chemical composition. Further, pollutant compounds have distinct properties while reacting with different environmental compartments. Some compounds have an affinity to the water phase and the solid, while others are more attracted to either the solid or the liquid phase, and others are highly volatile. The fate of antibiotic compounds in the environment is impacted by seasonality, organic and inorganic nutrient accumulation, compound transformation, oxygen demand in the water and underlying sediment, presence of bacteria, algal and pathogen growth, climate, and flow variabilities. These biological, environmental, chemical, and weather factors determine their destination and fate [20–23]. Hence, the system representation of the dynamic pollutant degradation mechanisms has evolved into complex modeling to reduce prediction uncertainties.

Several mathematical models have been developed worldwide to simulate the fate of antibiotics in the aquatic environment. Most of these models are process-based and use a set of mathematical formulations to combine existing theories, principles, and empirical knowledge of physical, biological, chemical, and ecological phenomena and are composed of tools and systems to simulate the fate of antibiotics realistically [24]. The use of these models necessitates a broad and interdisciplinary understanding of their simulation mechanism, parameter estimation, data requirement, and validation, as well as strengths and limitations, to know the applicability and prediction uncertainties.

This study critically reviews the source, natural attenuation mechanism of frequently detected antibiotic groups, and the state-of-the-art on development and applicability of environmental fate and transport models. The paper critically discusses the strength and drawbacks of fate and transport models that are used to simulate the transport of antibiotics in the natural surface water medium. An effort has been made to evaluate the models' efficiency in their application to different environmental settings across the globe. The study aims to critically evaluate current model development processes and identify gaps in the reviewed models for further research to provide future perspectives.

## 2. Source of Antibiotic Contaminants

Antibiotics are considered a critical emerging contaminant in aquatic environments [25]. In order to understand the fate and transport of antibiotic contamination, the primary step is to identify their origins. Aquatic ecosystems receive contaminants in two ways: diffuse/nonpoint and point sources. Point source pollutants are pollution from a particular contaminant source having a specific discharge point. Point sources include pollutant discharges from Waste Water Treatment Plants (WWTPs), industries, commercial outlets, urban drainage, hospitals, and market. Nonpoint source pollutants are pollution from diffuse sources, which do not have a defined discharge pattern and are distributed over an area. These areas include pollutants from agricultural areas, informal settlements, animal husbandry, fish cultivation, pastoral grazing areas, and sewer leakages.

There has been a high dependence on antibiotics for human and veterinary applications. The average annual consumption rate of pharmaceuticals is estimated to have increased yearly by 5% since 2017 [20]. Antibiotic consumption in 76 countries (Asia, Europe, and North America) from 2000 to 2015 was reported to be 21.1 to 34.8 billion in defined daily doses, with an estimated 200% increase rate by 2030 [26]. Due to incomplete metabolization, approximately 50–80% of antibiotics administered to humans and animals are excreted into the environment as urine and feces and transported with wastewater [27]. WWTPs generate the major antibiotic residual contribution to the aquatic environment [20]. WWTPs receive untreated wastewater from various sources, such as households, commercial outlets, hospitals, and factories. Currently, most WWTPs are operated using conventional treatment techniques, which are less effective in treating organic compounds. Although advanced removal techniques, bioreactors, advanced oxidation, and nanotechnologies are applied, WWTPs cannot completely remove all residues. As a result, high antibiotic residue concentrations are detected in WWTP effluents [28–32]. Therefore, WWTP sludge and effluents are known to be a significant source of antibiotic pollution in the aquatic environment. Wastewater and sludge generated from pharmaceutical industries contribute to high concentrations of antibiotic contamination in aquatic environments [33]. Further, household usage of antibiotics, unused and patient-utilized antibiotics from hospitals, and unintentional disposal due to the lack of an engineered solid waste collection system are other potential sources for extensive antibiotic release to the environment. Solid waste disposal sites accumulate unused prescriptions from municipal waste, creating diffuse antibiotic pollution from wash-off due to intense rainfall, and are transported into the water environment through surface runoff and infiltration. Furthermore, due to sewer leakage and lack of connection to sewer systems, wastewater from households and industry are disposed of into the river system directly or through runoff and contributes to significant contamination in the environment.

Additionally, food-processing industries, aquaculture, and livestock farming, generate considerable residual antibiotic contamination [34]. Antibiotics applied for breeding, growth promotion, and infection treatment in aquaculture and livestock farming account for 63–84% of total usage [33]. In livestock, unabsorbed antibiotic residues are excreted as feces and urine, while in aquaculture, antibiotics applied through direct feeding and splashing produce extensive waste directly entering the water environment.

## 3. Antibiotic Degradation and Fate Kinetics in Surface Water

Given their high concentration in wastewater, antibiotics have been frequently detected in surface waters [35]. However, the concentrations of antimicrobials in surface water are lower than in wastewater due to degradation (e.g., uptake and metabolism by aquatic microorganisms, photolysis due to sunlight exposure, and hydrolysis). The concentration of antibiotics in the water and sediment occurs as freely dissolved organic matter bound (antibiotics bind to dissolved organic matter) and particulate solid bound (antibiotics bound to solid) [36]. Antibiotics in the dissolved form are transported in the water flow, whereas antibiotics in the bound form are suspended and transported by sediment (particulate organic carbon). The fate and persistence of antibiotics in natural waters are subjected

to various instream reactions as they travel downstream. It attenuates, degrades, decomposes, and transforms into another compound during transportation. The typical instream mechanisms that decrease the aqueous concentration of antibiotics in the water column may be categorized as physical (sorption, resuspension, sedimentation, and volatilization), biological (bioconcentration, uptake/co-metabolism, and biodegradation), and chemical processes (photodegradation, oxidation, and hydrolysis).

The effect of reaction processes on a particular antibiotic differs and depends on various factors. For instance, the sorption process is more likely to rely on the geological composition of the medium through which the river flows and the organic carbon content of the suspended solid and underlying sediment. Hence, higher quantities of organic carbon content in underlying sediments could result in a higher sorption rate of antibiotics depending on their physiochemical properties [37]. Further, the hydrolysis of antibiotics is sensitive to the system's ionic strength and pH value. The cationic forms of antibiotics are more susceptible to hydrolysis than neutral and anionic forms [37]. The density of microbes and the presence of high carbon content in the surface water play an increased rate of antibiotic biodegradation. The presence of organic carbon in the river water is limited to sediment, sludge, and wastewater, resulting in a lower biodegradation rate of antibiotics in the water column [37]. Sunlight exposure and the presence of sensitive substances favor a higher rate of photolysis. Therefore, fate modeling of antibiotics in the environment requires a particular investigation of specific rate kinetics.

Identifying possible classes of antibiotics detected from specific sources can be helpful for a better understanding of their persistence and evolution in the water environment. The most frequently detected antibiotic classes in natural water channels are sulfonamides (sulfadiazine, sulfadimidine, sulfamethoxazole), Quinolones (fluoroquinolones, norfloxacin, ofloxacin, ciprofloxacin, enrofloxacin), Beta Lactams (penicillin, ampicillin, Carbapenems, carbamazepine), tetracyclines (tetracycline, oxytetracycline, chlortetracycline, and doxycycline) and macrolides (erythromycin, Roxithromycin) [27,38].

The fate of specific antibiotics may not be equally affected by all the instream degradation reaction mechanisms, as these reactions are influenced by their reactive functional ring. For instance, in a natural river, transport of the tetracycline group of antibiotics has a higher rate of sorbing to sediments than other antibiotic groups [27,35]. Sulfonamides are highly susceptible to biological degradation, while biotic degradation has a minor role in tetracyclines [39]. Fluoroquinolones are sensitive to photolysis and are reported as the primary removal mechanism, resulting in a high photodegradation rate [40]. Microbial communities readily biodegrade macrolides, while photodegradation has a minor role in macrolide degradation due to its insufficient chromophore groups to absorb sunlight energy [33].

Antibiotics can be released into the aquatic system in altered and unaltered metabolic forms [41]. The altered metabolites may be reversed back to the parent compound due to hysteresis. Further, significant amounts of transformed toxic compounds are frequently released from WWTP effluents [38] and transformed by-products of some compounds may become more toxic and persistent than the parent compound, increasing the concentration of antibiotics in the water column [42,43].

The instream degradation mechanism of antibiotics is dynamic due to many interrelated participant elements. Specific consideration of degradation mechanisms will help to estimate the elimination rate of a particular antibiotic accurately. These dynamic processes jointly determine the accurate residual concentration level and degradation rate of the antibiotics in the aquatic medium. Therefore, it is crucial to understand the effect and dynamics of the elimination processes of a particular antibiotic.

### 3.1. Biodegradation

Biolysis is the biological degradation of antibiotics caused by enzyme-mediated transformation, primarily due to bacteria, human and animal metabolism, and microorganisms, such as fungi, microalgae, and protozoa [44]. The biological removal of antibiotics by an organism is through three forms: (1) through modifying the antibiotics compound

(biotransformation); (2) by cleaving the antibiotics (biodegradation); and (3) through mineralizing the compound (subsistence) [39]. Generally, in terms of removal mechanism, biodegradation encompasses distinct processes, including mineralization (conversion of an antibiotic compound to an inorganic by-product), detoxication (alteration of antimicrobial substance to a harmless product), and co-metabolism (the simultaneous metabolism of two compounds that act as substrates to each other). Other transformation processes include activation (transforming a non-toxic compound to a toxic compound by microbial action) and defusing (converting the antibiotic compound to a harmless metabolite before its potential is realized). Uptake and metabolism in humans, animals, and microorganisms may degrade or transform antibiotic compounds into other byproducts. Some microorganisms degrade antibiotics as they endure, using substances, including antibiotics, as sources of carbon and energy sources [39]. Therefore, the intracellular and extracellular enzymes of bacteria and fungi partially or fully cleave the ring in the compound and transform it into other substances.

Bacteria, such as phyla Proteobacteria, Actinobacteria, Bacteroidetes, Gammaproteobacteria, and fungi, such as white-rot fungi, Trametes versicolor, Pycnoporus, Tremellomycetes, and Cerrena unicolor, play a significant role in metabolizing various antibiotics [39]. For instance, white-rot fungi secrete lignin peroxidase, manganese peroxidase, and laccase which degrade 58 to 78 % of tetracyclines in 15 days [27,45]. In the presence of nitrogen, microorganisms transform sulfamethoxazole into 3-amino-5-methylisoxazole, which lacks antibiotic activity [39]. Bio enzymes initiate degradation through demethylation and N-oxidation of macrolactone ring and sugar loss of macrolides (azithromycin, erythromycin, and clarithromycin) [33,46]. Further, bacterial populations that are resistant to antibiotics can degrade antibiotic concentrations. For instance, the resistant bacterial population removes 40% of recalcitrant ciprofloxacin over 104 days in a natural river [40].

Microorganisms in the water and sediment–water interface partially or fully decompose degradable antibiotics into non-toxic compounds, inorganic by-products, and metabolites. Depending on the compound's chemical structure and the degrader type, compounds can be biodegraded into stable solutes. Further, the microbial-mediated transformation of antibiotics may cause the parent compound to detoxify, mineralize, or activate potential toxins. For instance, in the presence of nitrate as a nitrogen source, Gammaproteobacteria metabolize sulfamethoxazole into 4-nitro-sulfamethoxazole and des amino-sulfamethoxazole. However, 4-nitro-sulfamethoxazole is a more toxic compound than the parent compound, sulfamethoxazole. Consequently, upon the total consumption of available nitrate, it reverts to the parent compound form [39]. In such circumstances, only minor molecular change occurs with no loss of antibiotic activity [47]. Therefore, particular consideration should be set to such biotransformation processes when evaluating the removal rates of a specific antibiotic.

Biological degradation of antibiotics is influenced by various factors, such as the antibiotic's physical and chemical properties (degradability and concentration of the compound), environmental parameters (sediment carbon content, retention time, and seasonality), microbial metabolism (density, growth, and the type of microorganisms), and water quality (dissolved oxygen concentration, PH, nutrient level and temperature). A higher concentration of antibiotics constrains microbial activity. Low-antibiotic concentrated water has a higher biodegradability rate than high-concentration antibiotics [27].

The ambient temperature of the water plays a significant role in antibiotic transformation and biodegradation rate. A limitation of nitrifying microbes due to low water temperature is hypothesized to cause poor antibiotic removal in rivers [48]. On the other hand, anoxic (denitrifying) conditions cause reversible reactions and promote prolonged antibiotic persistence [49]. Higher temperatures initiate microbial physiological activity, enzyme-dependent reaction, and antibiotic molecules, consequently increasing the biodegradability of the antibiotics. The biodegradation of sulfamethoxazole has a half-life of 77 days at 4 °C in natural river water, five times higher than that at 25 °C (16 days) [41].

Therefore, particular consideration of these environmental factors in determining the biodegradation rate would help in the realistic prediction of models.

### 3.2. Hydrolysis

Hydrolysis involves the oxidation of a compound by enzymes and the reaction of a compound with either the hydrogen or the hydroxyl component of the water molecule producing hydrolyzed compound. The molecular structure of some antibiotics contains hydrolyzable functional groups (amide, carbamate, ester, and halogens), rendering antibiotics susceptible to hydrolytic degradation [50]. These hydrolyzable functional groups also appear frequently in many classes of antibiotic molecules (penicillins, cephalosporins, and macrolides). Hydrophilic contaminates containing polar ions will have a higher tendency for hydrolysis. Hydrolysis is the primary removal mechanism in most polar or hydrophilic antibiotics, especially for amid and ester-containing substances, such as macrolides and β-lactams.

The rate of hydrolysis is controlled by the structure of specific antibiotics or reactants, the pH of the water, and temperature [51,52]. For instance, amoxicillin, one of the β-lactams antibiotics, easily degrades into amoxicillin penicilloic acid, amoxicillin 2′,5′-diketopiperazine, amoxicillin penilloic acid, and 3-(4-hydroxyphenyl) pyrazinol due to the hydrolysis instability of its β-lactam ring [53]. Penicillin is hydrolyzed to penicilloic acid by the activity of β-lactamase enzyme [44]. Conversely, fluoroquinolone and sulfonamide antibiotics hardly break down via hydrolysis under normal environmental conditions. These antibiotics lack structural features that are easily hydrolyzed under environmental circumstances.

At a high temperature, sulfonamides are ideal for hydrolysis and hydrolyzed to sulfanilic acid, attributed to cationic forms of sulfonamides in acidic solution, which are more sensitive to hydrolysis than the neutral and anionic forms of the compound. The hydrolysis of fluoroquinolones requires higher temperatures and higher concentrations of bases in the environment, which rarely occurs. Further, temperature variation in the same compound results in a variable hydrolysis rate constant. In an ambient environmental condition (pH 7 and 25 °C), the hydrolysis half-lives of cefalotin, cefoxitin, and ampicillin were 5.3, 9.3, and 27 d, respectively. However, with an increase in temperature (pH 7 and 60 °C), the hydrolysis half-lives of cefalotin, cefoxitin, and ampicillin were 0.067, 0.11, and 1.1 d [51]. A natural river sample with 100 µg/L amoxicillin solution lost 90% of amoxicillin in 8 weeks at 4 °C, while becoming completely lost at 20 °C in the observed time [52]. The hydrolysis of cefalotin, cefoxitin, and ampicillin at pH 4 was negligible, while the rate of their hydrolysis was found to be rapid in alkaline conditions (the half-lives at pH 9 and 25 °C were 1.4, 6.6, and 6.7 d, respectively) [51].

### 3.3. Photodegradation

Photodegradation is the abiotic degradation of compounds caused by sunlight absorption. Photolysis is the degradation pathway for most antibiotics in natural water and wastewater treatment [54,55]. Photodegradation is the primary elimination pathway for fluoroquinolones, sulfonamides, and tetracyclines [40,56]. There are two forms of photodegradation: direct and indirect (reaction due to photochemical-produced reactive intermediates). Direct photodegradation is the chemical degradation of a compound due to the direct absorption of solar radiation, leading to the formation of highly reactive intermediates or radicals that can cause the compound to break down into less complex molecules [57]. Indirect photodegradation occurs when photochemically produced reactive oxidant species (singlet oxygen ($O_2$), superoxide ($O_2^-$), hydrogen peroxide ($H_2O_2$), peroxyl radicals (OOR), triplet excited state dissolved organic matter (DOM), and hydroxyl radicals (OH)) transform compounds into a molecule [56,58]. Photosensitizers photochemically produce the reactive oxidant species under light exposure. A series of intermediate (reactive oxidant) species could be generated when light absorption transforms a substance surrounding the antibiotics into an excited state. Consequently, the active species

reacts with the antibiotic and initiates a faster decomposition by shortening the half-life of residual antibiotics [59].

The photolysis removal of antibiotic residues in the environment is influenced by various factors, such as water composition (such as organic and inorganic compounds, type and content of dissolved organic matter), water property (conductivity, pH, and temperature), photo intensity, structure and property of organic pollutants, the functional group of the antibiotics and algal biomass. The photolysis mechanism is primarily influenced by latitude in addition to cloud cover, a fraction of daylight, water level (depth), and bacterial and algal biomass.

Algal and bacterial biomass secrete photoactive extracellular and intracellular organic matter. The degradation rate of antibiotics increases when the radical photocatalyst species grow due to the high concentration of algal release of organic matter [57]. However, the produced organic matter may either induce the generation of photosensitizers (active species, such as chlorophyll, protein, humic, and fulvic dissolved organic matter) upon light absorption [59], where they may improve the photolysis rate or absorb irradiation by masking the light and scavenging of reactive oxygen species, playing a decisive role in the photolysis rate of antibiotics. Many studies report the effectiveness of the algal-mediated photodegradation of antibiotic residues [57,59,60]. Tian et al. [61] reported a 90% removal of chlortetracycline by algal-induced extracellular organic matter, while [57] reported a 38% elimination of ofloxacin by algal matter-induced photolysis. Increasing the concentration of organic matter (fluvic acid) from 5.0 to 50.0 mg/L under a simulated sunlight irradiation of light wavelength of 300–800 nm and temperature 21 °C decreased the rate constant of sulfamethoxazole from $3.0 \times 10^{-3}$ to $1.0 \times 10^{-3}$ min$^{-1}$ [62], attributed to the scavenging of oxidant species. Thus, photo-reactive intermediates produced by DOMs trigger the removal of most human antibiotics in the aquatic environment by indirect photodegradation [58]. However, the composition of DOMs influences the degradation of a particular antibiotic. For instance, the DOM with an oxygenated environment increases the degradation rate of sulfathiazole, while the steady purge of $N_2$ gas (deoxygenated condition) decreases the degradation of sulfamerazine [58].

The pH of a solution changes the photolytic forms of organic matter and antibiotics, as well as the rate at which reactive intermediates are generated. Additionally, the pH variation in water changes the ionic strength of pollutants and affects the electron transfer capacity and their reaction activities [56]. The change in pH plays a decisive and positive role in the photolysis removal of antibiotics. Tetracyclines are relatively stable in acidic conditions and unstable in alkaline conditions, attributed to the inter and intramolecular proton transfer in excited states [39]. Fluoroquinolones can exist in various pH forms (cationic, zwitterionic, or anionic form), affecting their direct photolysis rate. Under sunlight, the rapid removal of fluoroquinolones occurs in neutral and slightly alkaline conditions. Minimal antibiotic degradation rate kinetics from direct photolysis and volatilization is evident during cold seasons, especially in winter, due to low water temperatures and cloud cover [63]. This is attributed to low organic matter production from bacterial and algal biomass and limited photochemical generation of reactive species from photosensitizers. Deep water bodies have limited temperature penetration throughout the water level to the bed material. As a result, antibiotics in deep water bodies have decreased incidences of photodegradation.

The presence of nitrate ion concentration inhibits the indirect photodegradation of sulfathiazole, while dihydrogen carbonate enhances the photodegradation of sulfathiazole [58]. Nitrate ion concentration does not affect sulfamerazine degradation, while hydrogen carbonate inhibits the photodegradation of sulfamerazine [58] and promotes the degradation of sulfisoxazole [56]. The presence of halogen ions ($Cl^-$ and $Br^-$) inhibits the degradation of most sulfonamide antibiotics [56]. Therefore, understanding the natural removal pathway and establishing a reliable estimation mechanism for an accurate photodegradation rate is critical for the environmental fate modeling of antibiotics.

*3.4. Sorption*

Sorption is the attachment of dissolved substances from the aqueous to the solid phase. Sorption involves accumulating dissolved substances on solid surfaces by adsorption and the penetration or intermingling of substances with solids by absorption. Numerous studies report the high concentration of antibiotics in stream beds and suspended matters, considering that sediments act as sinks of antibiotics [64–68]. Sorption is vital in determining the reactivity, mobility, persistence, volatilization, and bioavailability of pollutants in natural waters, as it controls the antibiotic concentration between the water, suspended solids, or sediments [69].

The sorption of antibiotics in the water column follows a pseudo-first-order reaction. The water–sediment partitioning coefficient is used to determine the sorption capacity of the solid [25]. The sorption of antibiotics onto the suspended solid and the complexation of dissolved organic matter are determined by the octanol-water portioning constant ($K_{ow}$) and the fraction of organic carbon ($f_{oc}$) as presented in Equations (1)–(5) [63]:

$$\frac{dC_b}{dt} = \frac{Ck_{dom}DOM}{1 + k_{dom}DOM + k_{dss}TSS} \tag{1}$$

$$\frac{dC_p}{dt} = \frac{Ck_{dss}TSS}{1 + k_{dom}DOM + k_{dss}TSS} \tag{2}$$

$$k_{dss} = f_{oc}K_{oc} \tag{3}$$

$$k_{dom} = f_{oc}K_{oc} \tag{4}$$

$$LogKoc = 1.18logKow - 1.56 \tag{5}$$

where $C$ is the concentration of the antibiotics (ng/L), $C_b$ is dissolved organic matter bounded antibiotic concentration (ng/L), $C_P$ is the particulate solid bound antibiotic concentration (ng/L), t is time (day), $K_{dom}$ is the complexation coefficient between antibiotics and dissolved organic matter (L/kg), and $K_{dss}$ is the partition coefficient between antibiotics and solid (L/kg).

The exchange in the hyporheic zone of river channels is an essential medium in removing antibiotics from natural water bodies. Depletion of compounds in the water and sediment creates a concentration gradient (equilibrium imbalance) between the concentration of the aqueous and bounded form. Consequently, the adsorption and desorption processes rule out the migration of the antibiotics from one phase to the other. Residual antibiotics may sorb onto the solids by adsorption or settle in the sediments. Further, it may diffuse from the benthos' solid-bound phase to the water column's dissolved phase or be absorbed into the solids of the underlying benthos. Antibiotics in the sediment and sediment–water interface can be depleted by biotic activity (biotic depletion), facilitating resuspension and desorption, which cause the water column to be continuously polluted by migrating the antibiotics from the solid phase to the water column. For instance, biotic depletion of sulfamethoxazole in the benthos in the sediment–water interface made a concentration gradient and is absorbed into the sediment, leaving the water column [41]. Conversely, the depletion of sulfamethoxazole in the water column suggests the desorption of sulfamethoxazole from the sediment into the water [70]. As a result, the underlying sediment in the aquatic environment causes the prolonged presence of antibiotics [33]. Hence, sediments are regarded as secondary sources of diffuse pollution in water columns, releasing adsorbed residue to the overlying waters.

Sediments are porous matter containing loose particle spacing, enabling them to have many sorption sites on their surfaces. As a result, antibiotics bind to the surface of the sediment particles via electron attraction induced by their functional group. The adsorption of antibiotics is influenced mainly by their physiochemical properties, the

composition of the adsorbent matrices, and the state of the surrounding environmental element. Antibiotics with amino, hydroxyl, and carboxyl functional groups adsorbed to solid/sediment by cation exchange, hydrogen bonding, cation bridging, and other mechanisms [69,71–73]. For instance, cation exchange and hydrophobic partition play a vital role in the sorption of fluoroquinolone and sulfonamides. However, this does not necessarily mean other mechanisms do not affect their sorption. The mechanisms depend on and are influenced by the sediment's pH, ionic strength, temperature, organic matter, type, and size. The high-level organic carbon content in the sediment enhances sorption capacity [37]. Finer particles, such as silt, clay, and organic matter/detritus, are the ideal sorbents to have a greater sorption capacity and transport sorbate due to their large surface area-to-volume ratio.

Antibiotics carry polar and charged ions throughout the environment, which are influenced by pH [10,65,69,71]. The polar and charged ion interacts with the surface charge of the solid. Antibiotics with a positively charged functional group have an affinity to interact with negatively charged sediment via cation bridging. Antibiotics with the polar functional group may interact with the acidic group of sediment by hydrogen bonding. Overall, the opposing charges of the contaminant and the environmental medium facilitate antibiotic adsorption. Tetracycline's ammonium group interacts with negatively charged sites via cation exchange, while the negatively charged tricarbonyl methane keto–enol moiety (tetracycline derivative) interacts with negatively charged sites through cation bridging [69]. Furthermore, a fraction of the sediment sorbed may desorb due to environmental changes (salinity, pH, and temperature), resulting in hysteresis. This reversal process is highly dependent on the energy of the chemical bond between the sorbate and binding sites and thus affects the antibiotic sorption rate [71].

The availability and composition of different ions in the system will give the sorption capacity, and Kd values different degrees of change. The presence of metallic ions $Fe^{3+}$, $Na^+$, $K^+$, $Ca^{2+}$, and $Mg^{2+}$ in an acidic solution compete with Tylosin, an antibiotic from the erythromycin family, which decreases the adsorption capacity of the solid to macrolides [33,74]. In neutral and alkaline solutions, metallic ions act as proton donors and may promote sorption [33]. In most cases, the positively metallic minerals are attracted to negatively charged surfaces forming a metallic ion complex with functional groups of antibiotics, inhibiting the sorption process and playing the decisive role [71,72,75,76].

Most fluoroquinolones' affinity to sorption is controlled by cation exchange, as most of them are cationic species in natural pH and can be easily attracted to negatively charged solids or sediments [72]. Conversely, sulfonamides poorly adsorb into the sediments and are detected in relatively high concentrations in the pore water than the sediment [74]. This is attributed to the electrostatic repulsions of sulfonamides, primarily acidic chemicals, which adsorb poorly in alkaline conditions [74]. Thus, due to their weak adsorption into solids and sediments, sulfonamides pass the hyporheic zone and swiftly reach and contaminate groundwater [39].

The change in pH affects the dissociation of antibiotics and the surface charge of the solid [71]. The pH is the primary factor affecting the sorption process. In most cases, the sorption rate of antibiotics decreases with the increase in pH. The sorption rate of chlortetracycline in natural water decreased from 0.68 to 0.54 at a pH of 7.9 and 9 [71]. The sorption rate of tetracycline decreases linearly with an increase in pH from 7, 7.5, 8, and 8.5 [69]. Temperature is another factor that influences the sorption of antibiotic contaminants. Increased temperature beyond normal room temperature decreases sorption rates. The variation in water temperature influences the solubility of pollutants (the higher the temperature, the higher the solubility) and influences the dissociation accordingly. Temperature rise increases the solubility and reduces the pollutant's hydrophobicity, further reducing the antibiotics' adsorption on the sediment's surface.

### 3.5. Oxidation

Oxidation is a process where contaminants in water are oxidized into easily degradable and harmless products with the help of reagents such as ozone, hydrogen peroxides, and permanganates [3,77]. The reaction of oxidants and naturally occurring hydroxide anions ($H_2O_2$ occurs in rainwater, sea, and freshwater) in water could form hydroxyl radicals, promoting the oxidation of targeted antibiotics [78].

Different combinations of advanced oxidants in a simulated system have been implemented to oxidize recalcitrant compounds into easily degradable or non-harmful products. Ozonation with hydrogen peroxides is employed in wastewater treatment to eliminate persistent antibiotics, such as ibuprofen and fluoroquinolone. Ozonation with hydrogen peroxide removes 99% fluoroquinolones and ibuprofen in a 5-min reaction [3,78,79]. Alongside the naturally occurring hydroxides, synthetic substances discharged into the environment have an important influence on the oxidation of antibiotics [62,80]. Synthetic pollutants and disinfectants released into the environment may contain oxidizing reagents, anions, and cations. Hence, these oxidizing reagents in the natural water play a significant role in the oxidation process of antibiotics. For instance, anions, such as sulphate, chloride, nitrogen, and phosphorus, play an oxidation role by creating weak oxidants, such as chlorine, sulphate radical, and peroxydisulfate ion from hydroxyl radicals [3,79].

### 3.6. Bioaccumulation

Bioaccumulation is the build-up of antibiotics in the body of aquatic organisms, such as algae, plankton, daphniids, bivalve, benthic mollusk, and fish through the food chain. Aquatic biota take their food from suspended matter, sediments, and microscopic species in surrounding water [81,82]. Sediments and suspended matter (microscopic algae, bacteria, and detritus) are major sources for filter feeding and benthic organisms [82]. Metabolization and chemical transformation play a vital role in the bioaccumulation of antibiotics. Notably, the accumulation of antibiotics in different aquatic species has been reported by numerous studies [83–87]. The studies demonstrated the magnification of antibiotics in different tissues and organs of aquatic microorganisms. Therefore, ignorance of antibiotic bioaccumulation in biotic-inhabited water bodies (lakes) may not reflect the actual occurrence of antibiotic contamination.

### 3.7. Volatilization

A substance that has been in water can evaporate or escape into the atmosphere through a process called volatilization. Volatilization occurs due to the concentration difference between the dissolved concentration in water and the gas phase concentration in the overlying atmosphere in the interface of water and air. A compound in water volatilizes when the compound in water is oversaturated (the dissolved concentration of the compound in water is in an abundance of saturation dissolved concentration of the compound in water). The mass transfer rate of antibiotics depends on the properties of the compound and the characteristics of the waterbody and the atmosphere, including the molecular diffusion coefficient of the substance in the water and atmosphere, temperature, wind speed, current velocity, and water depth [88,89]. High vapor pressure, high diffusivity, and low gas solubility favor the volatilization of an antibiotic compound [88]. The volatilization of a compound is calculated by Hennery's law constant [89]. Due to the limited possible volatilization property of antibiotics; generally polar (log $K_{ow}$ mostly < 4.5) and not volatile (Henry constant, $K_H$, generally $< 10^{-3}$ Pa m$^3$ mol$^{-1}$) [90], volatilization has been rarely considered in process-based models and rarely studied in the fate and transport of antibiotics [91].

## 4. Antibiotic Fate Models

The complexity of mathematical simulations of antibiotics has grown as a result of the multitude of inclusion of physio-biochemical, hydrological, and hydrodynamic interactions [92]. One aspect is the systematic modeling of mixing in riverine transport.

The mixing in the riverine transport can be longitudinal, transverse, and vertical and is represented by one (1D), two (2D), or three (3D) dimensions depending on the nature and the purpose of the simulation required. Rivers with less storage, abstraction, and control are often studied using one-dimensional models under normal circumstances (natural flow), with the assumption that longitudinal dispersion plays a more significant role than vertical and traverse movement [2,24]. However, in regulated rivers and stagnant water, mixing is greatly influenced by various artificial regulations and abstractions (e.g., dams, sluices, canals, and storage), hydrodynamic processes, geometry, secondary flow, and aquatic organisms [24]. In stagnant water, mixing can be influenced by longitudinal dispersion, lateral mixing (diffusion), and vertical mixing and can be represented by 2D/3D modeling. As a result, realistic mixing representation grows more complex, progressing to 1D and 3D modeling.

The dynamic phenomena occurring in surface waters associated with the spread of various pollutants are often described using ordinary and partial differential equations. In a 1D to 3D process-based reactive modeling, the mass balance of a pollutant is generally expressed using the pollutant transport advection–dispersion equation (Equation (6)) [93]:

$$\frac{\partial C}{\partial t} = -u\frac{\partial C}{\partial x} - v\frac{\partial C}{\partial y} - w\frac{\partial C}{\partial z} + \frac{\partial}{\partial x}\left(D_x\frac{\partial C}{\partial x}\right) + \frac{\partial}{\partial y}\left(D_y\frac{\partial C}{\partial y}\right) + \frac{\partial}{\partial z}\left(D_z\frac{\partial C}{\partial z}\right) + K_t C + S \quad (6)$$

where $u$, $v$, and $w$ are the velocity components in the $x$, $y$, and $z$ coordinates in three dimensions; $Dx$, $Dy$, and $Dz$ are components of the dispersion coefficient; $S$ is an external source to the system due to loads, boundaries and descends, and $t$ is the time. $K_t$ is the total of all constants (Equation (7)) that account for all biotic and abiotic processes involved in the overall removal of antibiotics [94]. Very often, all the degradation processes are regarded to be first-order constants:

$$K_t = K_b + K_h + K_p + K_o + K_v + K_{bc} + K_s \quad (7)$$

where $K_b$, $K_h$, $K_p$, $K_o$, $K_v$, $K_{bc}$ and $K_s$ are the biolysis, hydrolysis, photolysis, oxidation, volatilization, bioconcentration, and sorption constants, respectively.

The 1D model is better suited to study catchment-scale runoffs and assess pollutant loadings based primarily on physical transport processes. A limitation of 1D models to simulate an ecological interaction paradigm with the surrounding environment is surmounted by coupling it with fugacity models. Multimedia fate models are good at simulating the distinct behavior of antibiotics in a multimedia environment [24]. However, consideration of exposure in a multimedia environment (air, soil, water, and sediment), the fugacity concept, may increase the level of complexities in modeling. Fugacity is the tendency of a compound to prefer one phase over the other in a similar pressure and temperature. This model includes partitioning concentration among multimedia, accounting transformation, and handling variabilities. In fugacity models, a different simulation technique comprises different combination processes. The level of accounting corresponds to Level I, considering equilibrium distribution among the compartments without transformations, and to Level IV, giving advection, transformation, hysteresis (antibiotic hydrolyzed back to the parent molecule), and variabilities (without equilibrium distribution between phases) [11]. Compared to 1D models, 2D/3D water quality models may better describe the fate, interactions, mixing, and transport of antibiotics within water bodies (e.g., reservoirs and lakes).

In each model, hydrology can be characterized in two ways: gridded approaches incorporating extensive process-based hydrological models, or segmenting the river network into discreet river segments with calibration against measured hydrologic data and exploration to ungauged sites [95]. Both approaches have different data requirements, processing efforts, and computational efficiency. In addition to using different hydrological characterization, probabilistic simulations are applied to account for temporal variability of concentration caused by flow variabilities. The probability distribution depicts the expected change in concentrations over time due to discharge fluctuations and uncertainty

in the input parameter. Moreover, the model coupling is a common practice in antibiotic transport simulation. Often models employ coupling of external hydrological, hydrodynamic, and fate emission models. The coupling is expected to increase the integrity of the model, save the time spent on developing the external coupler model, and allow for a better prediction [24].

Moreover, the simulation of antibiotics from a process-based reactive transport model is dependent on the instream reaction kinetics [63,65,96]. Therefore, modeling the fate of antibiotics in a river must represent the possible processes affecting the persistence of the antibiotics. However, the physical representation of the transport process has dynamic, distinct, and complex properties, making developing a precise quantification mechanism a tough job. Ease of complexity in the expression of transport kinetics and parameter dependencies is possible through user customization flexibility in the models. A hybrid modeling framework that combines laboratory kinetic results with numerical modeling may help better understand the instream removal reactions effectively [24]. Therefore, the validity of the simulated concentration of antibiotics from these hybrid models can be assessed by correlating the simulated concentration with actual field monitoring data [17,97]. Validated prediction from the models is used for risk characterization to estimate the likelihood of the adverse effect due to exposure to predicted concentration.

Generally, the assessment and monitoring practice of antibiotic contamination in the environment requires selecting an appropriate fate model, parameter estimation, spatial resolution, emission, and fate estimation mechanism. The selection of a suitable model requires consideration of various factors depending on the level and purpose of the assessment. For instance, onsite wastewater treatment and direct discharge are not considered in some models. In most cases, antibiotic fate models have a stationary emission calculation and fail to consider the bypass flow from the WWTP. On the other hand, different models consider different instream decay mechanisms differently and have different parameter estimations. More notably, most of the models use spatially coarse-resolution river flow data. Hence, because different models use river flow data with varying resolutions, flow computation in the models would introduce significant prediction uncertainties.

A detailed review of the base formulation, applicability, and ease of use of commonly used antibiotic fate models is presented in the following section. The selected models focus on local, regional, and global antibiotic fate modeling. However, fate and transport models that have not been developed explicitly for antibiotics but have direct applications are included. The models are chosen based on their popularity and use in various regions (e.g., Asia, Europe, and North America). The review includes 11 standalone and web-based models, including Pharmaceutical Assessment and Transport Evaluation model (phATE), Geography referenced Regional Exposure Assessment Tool for European Rivers (GREAT-ER), Corps of Engineers Quality 2-Enhanced (QUAL-2E), Corps of Engineers Quality 2K, the 2K represents the period when the model was developed, which is the year 2000 (QUAL-2K), Global FATE, Water Quality Analysis Simulation Program (WASP), AQUAtic SIMulation Model (AQUASIM), in Silico Tool for the Risk Assessment of Antibiotics in the Environment with the focus on their Environmental Metabolites (iSTREEM), Quantitative Water–Air Sediment Interaction (QWASI), Exposure to pharmaceuticals in the environment (ePiE), and European Union System for Evaluation of Substances (EUSES). Their environmental applicability, sources simulated, simulated pollutants, and other aspects of the selected representative models are summarized in Table 1.

**Table 1.** Review of environmental antibiotic fate simulation models.

| Model | Environment (Medium) | Type (Sources Simulated) | Pollutant Modeled (Risk Assessment) | Advantage | Limitation | Open-Source User Interface | References |
|---|---|---|---|---|---|---|---|
| phATE | Streams Rivers Lakes Reservoirs | 1D (point) | Pharmaceuticals (antibiotics) (screen-level risk assessment) | Provides a range of load emission scenarios. | Highly sensitive to WWTP removal efficiency, affecting all predictions in the catchment. Does not consider in-sewer removal. Limited geographic scope. Coarse-resolution descript segment (~16 km). | Available | [98] |
| GREATER | Rivers | 1D (point and diffuse) | Pharmaceuticals (antibiotics) and nutrients (screen-level risk assessment) | It enables the study of potential risk management scenarios. It provides a statistical distribution of pollutants. The stochastic simulation enables to account for uncertainties in the input data. Efficient emission/source calculation. Considers removal of a compound during sewer transport. | The laborious pre-processing steps to set up the database and fill in the required data for the parameters. Calculates spatially steady-state concentrations susceptible to temporal fluctuation. Emission pattern calculation influenced by the WWTP bypass flow. | Available | [17,97,98] |
| QUAL-2E | Streams Rivers | 1D (point and diffuse) | Dissolved oxygen, organic nutrients, algal concentration, antibiotics | Simulation point and nonpoint sources. Provides simulation of non-uniform flow. | Cannot model the temporal variability of flow. The model gives a good simulation of narrow rivers (highly sensitive to water depth) as deep rivers have different stratification and mixing rate. It cannot simulate the effect of toxic organic compounds and heavy metals. It is inappropriate for waterbodies exhibiting significant lateral variations. | Available | [99,100] |
| QUAL-2K | Streams Rivers | 1D (point and diffuse) | Pharmaceuticals (antibiotics), conventional parameters | Enables to divide the river into unevenly spaced segments. Simulation of the effect of the generic pathogen, total inorganic carbon, and light extinction. | It is inappropriate for waterbodies exhibiting significant lateral variations. Did not consider the effect of sedimentation. | Available | [98,101] |
| Global FATE | Rivers Lakes Reservoirs | 2D/3D (point) | Pharmaceuticals (antibiotics) | Efficient fine resolution to represent small streams. Worldwide geographic scope | Cannot simulate flow variabilities. Require extensive data and external hydrological pre-processing steps. | Available | [102] |
| WASP | Rivers Reservoirs Lakes Estuaries Coastal areas Wetlands | 1D/2D/3D (point and diffuse) | Pharmaceuticals (antibiotics), conventional parameters | It enables analysis of the significance of individual mechanisms. It includes a sediment diagenesis module for remineralization. It provides a sensitivity analysis. | Difficult to obtain segment/site-specific data to calibrate fate mechanisms. It has a limitation in modeling concentration gradients in the mixing zone for wide channels with poor mixing conditions. Cannot simulate high-flow events. It requires external hydrodynamic models for flow information. | Available | [2,63,86,94,103] |

**Table 1.** *Cont.*

| Model | Environment (Medium) | Type (Sources Simulated) | Pollutant Modeled (Risk Assessment) | Advantage | Limitation | Open-Source User Interface | References |
|---|---|---|---|---|---|---|---|
| AQUASIM | Streams Rivers Lakes Reservoirs | 2D/3D (point) | Pharmaceuticals (antibiotics), conventional parameters | Efficient vertical mixing representation and temperature profiling. | It assumes uniform horizontal mixing in lakes and reservoirs. | Available on request | [104–106] |
| iSTREEM | Streams Rivers | 1D (point) | Pharmaceuticals (antibiotics) (conservative risk assessment) | Provides simple simplicity of simulation. | Suitable for simple simulation. Requires pre-processed data. Does not consider in-sewer removal. | Available | [107,108] |
| QWASI | Lakes | Multimedia fugacity (air, sediment, and water) (point and diffuse) | Pharmaceuticals (antibiotics) and organic pollutants (screening-level risk assessment) | Efficient and advanced modeling of lake temperature stratification. Modeling of ice melt in lakes. | Result influenced by choice and calculation of fugacity factor. Depends on uniform mixing condition. Requires exclusive half-life degradation data. | Available | [5,109–112] |
| ePiE | Streams Rivers Lakes Reservoirs Estuaries | 1D (Point) | Pharmaceuticals (antibiotics) | Ease of application. | Suitable for narrow rivers. It does not consider in-sewer removal. | Available on request | [95,113,114] |
| EUSES | Streams Rivers Marine | Multimedia fugacity (air, water, Sediment, soil, and groundwater) (point) | Organic chemicals, pharmaceuticals (antibiotics) (conservative risk assessment) | Simulation in multimedia, including groundwater pollution. Simulation of exposure through the food chain. Allows estimation of media-specific degradation | Provides steady-state concentration susceptible to temporal fluctuation. Extensive data requirement. Intensive pre-processing of model parametrization for site-specific simulation. | Available | [115–120] |

## 4.1. WASP

The Water Quality Analysis Simulation Program is a widely used surface water quality modeling framework used to address various pollution transport problems. It simulates flow in steady, unsteady, and non-uniform cases. WASP has the flexibility to model one, two, and three-dimensional problems. As a result, WASP has been widely used to simulate a wide range of pollutants [86]. One-dimensional simulation is commonly used to reduce data requirements and modeling complexity, assuming the significance of longitudinal dispersion over vertical and lateral mixing [2]. WASP uses the continuity (Equation (8)) and kinematic wave equation (Saint-Venant equations), Equation (9), to simulate one-dimensional flow and can perform in steady and unsteady flow conditions [2,98]:

$$\frac{\partial Q}{\partial X} + \frac{\partial A}{\partial t} = q(x) + K_t \tag{8}$$

$$\frac{1}{A}\frac{\partial Q}{\partial t} + \frac{1}{A}\frac{\partial}{\partial x}\left(\frac{Q^2}{A}\right) + g\frac{\partial y}{\partial x} - g\left(S_o - S_f\right) = 0 \tag{9}$$

where $A$ is the cross-sectional area of the system (m$^2$), $Q$ is the flow rate (m$^3$/s), $q(x)$ is the lateral inflows (m$^3$ s$^{-1}$ m$^{-1}$), $S_o$ is the bed slope of the channel, $S_f$ is the frictional slope, $y$, and $g$ is the water depth (m) and gravitational pull (m s$^{-2}$). The first and second terms in Equation (9) simulate the advection and convective process, respectively. The diffusion process (third, fourth, and fifth terms) are the pressure, gravitational and frictional forces. WASP uses the box approach to discretize the reach into segments (box). Each box is assumed to be thoroughly mixed, and the advection–dispersion transport equation shall apply [98]. Therefore, the model simulates advection, dispersion, point mass loading, diffuse mass loading, and steam transport processes through time variability [63]. Antibiotics in the suspended solid are also simulated through a solid transport module considering erosion, settling, and sedimentation. The model uses systematic stepwise fitting of transformation processes (biodegradation, hydrolysis, photolysis, and volatilization) to analyze the critical fate mechanism and enable rigorous analysis of the significance of individual mechanisms. However, the simulation requires specific data to drive the dependency of each mechanism. The unsteady simulation calibration and configuration of WASP require flow, concentrations, and hypergeometric properties entering the boundary segments. Further, it analyzes pollutant fate and transport problems in diverse water bodies, such as ponds, streams, lakes, rivers, reservoirs, estuaries, and coastal waters.

Besides the simulation of antibiotics, WASP has broadly been used in the simulation of conventional pollutants (e.g., nutrients, dissolved oxygen, sediment oxygen demand, eutrophication, algae, and bacterial contamination), toxic pollution (organic chemicals, metals, mercury, Pathogens, and organic chemicals), and persistent compounds [2].

## 4.2. GREAT-ER

Geography referenced Regional Exposure Assessment Tool for European Rivers is a steady-state model, with both deterministic and stochastic approaches (Monte Carlo simulation), approach developed for aquatic exposure prediction of antibiotics and customer products [9]. GREAT-ER uses Geographic Information Systems (GIS) and chemical models to simulate the fate of chemical compounds at the river basin level. Watersheds and hydrology data are managed using GIS. The model uses ArcView (ESRI) to store and visualize data and requires several parameters, including physiochemical, hydrological, consumption, use pattern, removal efficiency, and simulation data.

The essential elements of the model are antibiotic emissions from contamination sources and river segments [121]. Emission is calculated based on the population, water consumption, antibiotics consumption, extraction rate, and removal efficiency if WWTP is available [9]. The hydrology module comprises flow measurement from the gauging station and statistics of flow distribution through the catchment. Natural attenuation of antibiotics is described by first-order in-stream removal, assuming a fixed rate.

The reach of the river shall be discretized into segments, and each segment is attributed to flow, velocity, inflow, and outflow loadings. The river network is represented by a sequence of segments with a maximum length of 200 m [98]. Confluences, point emission sites, and regulation sites (monitoring sites, gauges, and weirs) are considered as nodes [122]. Pollutant loads from the WWTP are calculated using the plant's consumption and population data and the removal efficiency of the treatment type used. Each river stretch receives flow and quality data from the upstream segment and, if available, from tributaries and WWTP. Thus, the mass balance of loads and first-order instream loss processes shall be applied to each river segment to calculate the concentration in the segment [17].

GREAT-ER uses the Monte Carlo simulation to assess the distribution of predicted concentration and reflect the variability of various parameters (flow variabilities, emission rate, WWTP removal efficiency, and in-stream decay rate). GREAT-ER Monte Carlo simulation is widely used mainly for flow variabilities and WWT removal efficiency to account for their effect on the temporal variation of concentration of simulated antibiotics [17]. The model's stochastic simulation also enables the characterization of the input data to account for uncertainties. Furthermore, the model enables the study of potential risk management scenarios and the examination of the impact of built-in regulations on river water quality. It also provides the spatial distribution of the simulated concentrations in the catchment [97].

### 4.3. phATE

The Pharmaceutical Assessment and Transport Evaluation model is a deterministic model developed to simulate active pharmaceutical ingredients from point sources. The model divides the river into discrete segments. The phATE model is based on a mass balance (conservation of mass) equation describing the contaminant's fate in each segment [42]. Output from one river segment enters as an input to the subsequent segment and creates a series of sequences. Each river segment in the sequence receives pollutant load mass from the upstream segment and a WWTPs, if any [92]. Contaminants from a particular segment leave through first-order instream decay, abstractions, or flowing to the subsequent segment.

River segments are considered plug flow in reservoir modeling, while reservoirs are considered well-mixed tanks. Reservoirs are divided into segments of tanks and modeled as a series of tanks; therefore, mass concentration passes from one to the subsequent tank due to advection [98]. The mass load from a WWTP is calculated as the average annual compound per capita consumption multiplied by the total population served by the given WWTP. This mass load is then multiplied by two removal factors: loss factor by human metabolism and WWTP removal efficiency [92]. Then, all the relevant instream transport processes are considered by summing to a total first-order loss rate constant. The phATE model uses the Microsoft visual access database to store input and output data and uses GIS to manage watershed and hydrologic data. It also requires an external database or external model for hydrological inputs (such as mean and low-flow rates). The phATE model not only gives antibiotic simulation in surface water but it also used to estimate concentrations of active pharmaceutical ingredients in biosolids and sludge from WWTPs [19].

### 4.4. QUAL-2E (Q2E) and QUAL-2K (Q2K)

CE QUAL-2E is a hydrodynamic and water quality model developed by the US Environmental Protection Agency. It is widely used to simulate water quality parameters from point and nonpoint sources and abstractions [99]. QUAL-2E is a one-dimensional laterally averaged hydrodynamic model with a steady, non-uniform flow along the stream by computing a series of steady-state water surface profiles [123]. The channel is assumed well-mixed vertically and laterally. The reach of the river is divided into evenly spaced river segments, and water quality components are simulated on a diurnal time scale. Pollutant transport is simulated by solving the advection–dispersion equation based on the mass balance concept of transport and kinetic processes [101].

QUAL-2E simulates antibiotics in well-mixed narrow rivers and streams and is commonly used to simulate changes in point-source pollutant discharge and analyze the effects of nutrients on algal concentration and dissolved oxygen. QUAL-2E simulates traditional water quality parameters, generic conservative and non-conservative pollutants, and emerging contaminants, including antibiotics [98].

QUAL-2K is the updated version of Q2E with the improvement and addition of more water quality parameters and light extinction in calculating pollutant attenuation [123]. QUAL-2K provides an advantage to dividing the river into unevenly spaced segments. Multiple abstractions and mass loads can be simulated at any river segment in the reach [101]. Q2K includes a simulation of the effect of the generic pathogen, pH, total inorganic carbon, and light extinction on the simulation of a given antibiotic concentration.

### 4.5. GLOBAL-FATE

GLOBAL-FATE is a GIS environment-based fate and transport module that simulates the fate of human pharmaceutical compounds in the global river network, including lakes and reservoirs [124]. It solves the steady-state concentration of point source down-the-drain contaminants in the aquatic media (small streams, rivers, lakes, and reservoirs). GLOBAL FATE uses loss due to human metabolism, WWTP removal, dilution, and first-order mass decay as concentration attenuation mechanisms for human medicine [102]. The geographical and hydrological (including shape, location, and volume of lakes and reservoirs) and contaminant datasets are overlaid to compute the contaminant concentration [102]. Contaminant loads from the area are calculated using the population and per capita consumption. Contaminant mass load reaches the river network as an input load either directly or reduced by the factor of removal efficiency of the WWTP. The routing (flow accumulation and flow direction) in each river reach, or raster cell, is computed using the basic GIS module. Then, the attenuation mechanisms are sequentially applied throughout the river reach to compute the concentration of the contamination.

### 4.6. AQUASIM

The AQUASIM model was developed to simulate emerging contaminates (personal care products and pharmaceuticals, including antibiotics) and used to estimate kinetics parameters and uncertainties based on measured data [125]. The model solves partial and differential equations of the mass balance of pollutants using the implicit (backward difference), variable step, and variable order gear integration technique. The model represents the system by discretizing it into compartments. The compartment has four components. The mixed and partial mixing reactor components allow modeling uniform mixing, neglecting concentration gradients and differential mixing considering concentration gradients [104]. The river section component discretizes the river reach, while the reactor component models uniform mixing and the effect of algal and biofilm growth. During the modeling, the elimination mechanisms can be modeled based on user customization through the parameter estimation module of the model. The kinetic parameter estimation algorithm of the module allows users to explicitly calculate a system of equations to derive the degradation kinetics of a pollutant. It is used to model rivers, reservoirs, and lakes. Lake and reservoirs are modeled assuming varying vertical mixing through temperature profiles and uniform horizontal mixing [125].

### 4.7. QWASI

QWASI is a multimedia fate fugacity model developed in Canada to simulate the concentration of antibiotics, personal care products, and other organic pollutants in the air, soil, sediment, and water. The model is based on mass conservation, with mass balance equations established between environmental phases [126]. The model is designed mainly for the simulation of pollutants in lakes. QWASI is developed to simulate pollutant migration and transformation between air, water, and sediment phases by assuming a consistent level of water mixing in a lake [5].

The modeling is based on the lake's freeze–melt and temperature stratification characteristics due to temperature change. The model requires several parameters depending on the level of simulation: geographical and physical parameters (e.g., lake surface area, depth of sediment, suspended solid concentration and density, sediment burial rate, sedimentation, resuspension rate, mass transfer coefficient, mass fraction of organic carbon in sediment and water phase, sediment chemical oxygen demand, concentration and discharge rate of inflows to the lake), water quality (e.g., temperature, salinity) and antibiotic physical and chemical properties (e.g., water solubility, vapor pressure, Log $K_{ow}$, ice–water portioning coefficient, hydrolysis rate, photolysis, volatilization rate, melting point) [5,109–112].

The fate of chemicals is assessed based on the principle of fugacity, movement between different environmental mediums. Advection and diffusion are the major transport mechanisms in all mediums [127–129]. QWASI requires the user to have an explicit input half-life for the reactive degradation of antibiotics in each of the three-principal media (air, water, and sediment). QWASI model is also used to study environmental changes caused by temporary or permanent changes in environmental properties.

*4.8. iSTREEM*

iSTREEM is a web-based single medium contaminate fate assessment model developed in the US to simulate the fate of tracing contaminants, including antibiotics in WWTP effluents, water supply intakes, and receiving water bodies. iSTREEM provides a conservative estimation of down-the-drain chemicals under mean and low flow conditions in streams and rivers and allows a risk assessment of the estimated concentration. iSTREEM is based on previously developed ROUT and GIS-ROUT models [107]. The ROUT model is used to characterize WWTP loading, instream-loss, and properties of chemicals. GIS-ROUT allows the incorporation of special data (digitized river network) and enables spatial data analysis. It uses the per-capita consumption data to calculate WWTP effluent concentrations. iSTREEM is developed using visual basics and the primary function of ArcGIS to represent the river network and unique analysis of the reach. The river reach is divided into segments and chemical inputs including; upstream contributions, WWTP discharges, and losses due to instream degradation processes with dilution factors, should be applied to each segment using a first-order decay model. In iSTREEM antibiotics simulation, adsorption and biodegradation are assumed to be the main decay processes for the elimination of WWTP and the environment [130]. The model allows special variabilities and is flexible to customize to the geography of interest. Moreover, [131] used the iSTREEM algorithm with WorldPop, HydroBASINS, and HydroSHEDS databases to develop a special exposure assessment framework for Japan and China.

*4.9. ePiE*

Exposure to pharmaceuticals in the environment is a spatially explicit pharmaceutical model simulating the fate of contaminants in streams, rivers, and lakes at a special resolution of (~1 km). In ePiE, the river is discretized into a sequence of compartments and is connected by nodes that represent emission points, river junctions, and inlets and outlets of reservoirs, as well as lakes [95]. ePiE employs FLO1K GIS-based dataset for hydrological data. It also requires datasets to represent the location of nodes, lakes, and WWTPs. Often it uses HydroSHEDS, HydroLAKES, UWWTD-Water, and Hydro-BASINS databases to represent nodes, lakes, WWTPs, and basins, respectively. Gridded information on climate (air, temperature, wind speed), hydrology, and geochemical (soil, slope, chemical property) can be extracted for each node of the river network. Emission to nodes calculated using consumption and population applying reduction factor due to metabolism and WWTPs removal. Dilution and loss mechanisms in the surface water apply based on mass balance equations throughout the sequence of nodes. In ePiE, lakes and reservoirs are modeled as single, thoroughly mixed tanks (node) with additional hydraulic retention time and volume parameters.

*4.10. EUSES*

The European Union System for Evaluation of Substances is a multimedia fate model developed in the Netherlands and used to simulate point and nonpoint sources at a local, regional, and continental scale. EUSES uses partitioning of an area based on significance as an area fraction of water, the area fraction of agricultural soil, and industrial/urban soil. Those apportioned areas attributed by data of density of inhabitants, number of inhabitants and fraction connected to the sewer system, wastewater production, depth of water, temperature, precipitation, rainfall-runoff, wind speed, organic carbon content, suspended matter, sediment property, and particle content in the air [117]. Based on those data, concentration in each medium is calculated assuming various emission, distribution, transport, and removal scenarios. The model comprises emission, distribution, transport, effect, and risk characterization module [115,116,132].

The input module is used to feed primary data to the model, and the emission model is used to set emission factors based on the properties of substances. In the distribution route module, EUSES comprises two main models: The Simple Treat and Simple Box. The Simple Treat predicts the fate of chemical concertation for microorganisms through indirect emission. Simple Box calculates regional and continental environmental concentration through both direct and indirect emissions. The exposure module considers intake by humans and microorganisms. Therefore, exposure, effect, and risk assessment can be conducted for the environment and environmental population, including humans, microorganisms in sewage treatment plants, aquatic and terrestrial ecosystems, and predators.

Concentrations in water, suspended matter, and sediment can be calculated by considering partition coefficients, degradation rates, and volatilization. The concentration in water is calculated by presuming complete mixing and a constant dilution factor. In the soil model, the inputs are assumed from wet and dry deposition from air and sludge application and influenced by volatilization, degradation, and leaching, as well as the distribution with the partition coefficients. The degradation of compounds is accounted as a first-order constant based on the mass balance equation. EUSES can estimate media-specific degradation rates based on user input to characterize the substance susceptibility to biodegradation [118]. It is also used to compare the distribution of chemicals among air, water, and soil. Moreover, ref. [133] uses EUSES to compare the distribution of predicted concentrations of 40 chemicals and antibiotics among the environmental mediums, air, water sediment, and soil, and reported both over and underestimation depending on volatilization and sorption properties of a substance. EUSES is unsuitable for applying to a specific geographic interest [131].

## 5. Evaluation of Model Simulation Efficiency

The ability of a model to simulate the actual occurrence of antibiotic transport in water systems is evaluated through corroboration. Corroboration is the validation of a model prediction using monitoring data. This is achieved by comparing model simulations with observed (measured) data using different statistical metrics. A model with ease of use and closer simulation of existing reality is preferred. However, model simulation has a margin of error or uncertainty and is influenced by various factors.

For instance, significant discrepancies in the predicted concentration of antibiotics are possible due to different methods of specifying degradation rates [118]. Seasonal antibiotic human consumption and WWTP operation efficiency associated with degradation processes can bring prediction discrepancies. Climate, flow, and weather variabilities affect model prediction efficiency, as far as antibiotic fate and transport are concerned. As a result of a change in the hydrological condition, the concentration of pollutants may either increase due to the lack of dilution during dry periods or work in the opposite direction, facilitating removal (natural degradation processes) by increasing the residence time [49].

During flood events, different physical phenomena may occur at the same time: first, it may contribute to the overloading of WWTPs, thus reducing removal efficiency; second, is the remobilization of pollutants from sediments in the river; and third, it may wash

out pollutants from agricultural fields to the rivers. Therefore, incorporating seasonality (winter, spring, summer, and fall) parameters may help [42].

Different model prediction efficiency for different antibiotics has been observed in different countries (Table 2). The reviewed studies used different performance evaluation criteria. In addition to using different error metrics, most reviewed studies use graphical plot techniques to present the goodness of model prediction. Further, studies reported in GREAT-ER, WASP, and EUSES present plot comparisons due to their built-in plotting feature. One potential predictive performance is selected as their main statistical criterion for the studies that used more than one evaluation criterion. It is also important to know which model assessment criteria works best with each model [134]. Therefore, specific model evaluation criteria with the range of goodness of fit for each model are reviewed and presented in Table 2. For instance, correlation coefficient ($R^2$) is often preferred for GREAT-ER and EUSES [9,118,121,133,135], PBIAS for WASP [2], and RMSE for QUAL-2K [101]. The outcomes of the assessment criteria, on the other hand, may change in accordance with the number of observations. For an observation below 50, suitable evaluation criteria can be chosen by comparing the weightage and bias of a respective prediction assessment criteria [134].

**Table 2.** Representative model prediction performance evaluation of environmental antibiotic fate model simulation over a regional scale (country) application in different countries across the globe.

| Model | Target Antibiotics | Application | Error Metrics | Deviation | Often Used Metrics | Range | Region | References |
|---|---|---|---|---|---|---|---|---|
| phATE | Ibuprofen | Modeling concentration of antibiotics | $R^2$ | 0.48 | - | - | Canada | [42] |
| | Naproxen | Modeling concentration of antibiotics | $R^2$ | 0.68 | | | | |
| | Carbamazepine | Modeling concentration of antibiotics | $R^2$ | 0.68 | | | | |
| | Acetaminophen | Predict human antibiotics in surface water | MF | 47 | | | USA | [136] |
| | Erythromycin-H2O | Predict human antibiotics in surface water | MF | 4 | | | | |
| | Oxytetracycline | Predict human antibiotics in surface water | MF | 0.02 | | | | |
| | Sulfamethoxazole | Predict human antibiotics in surface water | MF | 9 | | | | |
| | Tetracycline | Predict human antibiotics in surface water | MF | 6.5 | | | | |
| GREAT-ER | Triclosan | Predict catchment concentration | MF | 0.769 | $R^2$ | ≥0.7 | Germany | [137] |
| | Triclosan | Predict local concentration | MF | 1.5 | | | | |
| | Bezafibrate | Predict local concentration | MF | 0.03 | | | Spain | [11] |
| | Carbamazepine | Predict local concentration | MF | 0.5 | | | | |
| | Citalopram | Predict local concentration | MF | 0.1 | | | | |
| | Diclofenac | Predict local concentration | MF | 0.1 | | | | |
| | Erythromycin | Predict local concentration | MF | 2 | | | | |
| | Fluoxetine | Predict local concentration | MF | 0.7 | | | | |
| | Ketoprofen | Predict local concentration | MF | 0.003 | | | | |
| | Trimethoprim | Predict local concentration | MF | 0.18 | | | | |
| | Atorvastatin | Predict regional concentration | MF | 0.2 | | | | |
| | Carbamazepine | Predict regional concentration | MF | 0.004 | | | | |
| | Fluoxetine | Predict regional concentration | MF | 0.52 | | | | |
| | Naproxen | Predict regional concentration | MF | 3 | | | | |
| | Trimethoprim | Predict regional concentration | MF | 0.08 | | | | |
| QUAL-2K | Diclofenac | Predict local concentration | RMSE | 0 to 80 | | | Spain | [121] |
| | Carbamazepine | Degradation study | RE | 5.85 to 6.82 | RMSE | ≤10% | China | [123] |
| | Triclosan | Degradation study | RE | −7.18 to −157 | | | | |
| WASP | Venlafaxine | Predict concentration in river water | PBIAS | −5 to −13 | PBIAS | ≤25% | Canada | [2] |
| | Naproxen | Predict concentration in river water | PBIAS | −1 to 5 | | | | |
| | Carbamazepine | Predict concentration in river water | PBIAS | −22 | | | | |
| | Venlafaxine | Predict concentration in river water | PBIAS | −9 to −26 | | | | |

**Table 2.** *Cont.*

| Model | Target Antibiotics | Application | Error Metrics | Deviation | Often Used Metrics | Range | Region | References |
|---|---|---|---|---|---|---|---|---|
| | Carbamazepine | Predict concentration in river water | PBIAS | −1 to −23 | | | | |
| | Venlafaxine | Predict concentration in river water | PBIAS | −28 | | | | |
| AQUASIM | Diclofenac | Degradation estimation of diclofenac | $R^2$ | 0.92 | _ | _ | Switzerland | [138] |
| iSTREEM | Carbamazepine | Modeling fate carbamazepine | MF | 0.5 | - | - | Canada | [108] |
| | Climbazole | Predict concentration in river water | MF | 4 | | | China | |
| QWASI | Amoxicillin | Antibiotic fate modeling in lakes | LGMF | 1.32 | - | - | China | [5] |
| | Ciprofloxacin | Antibiotic fate modeling in lakes | LGMF | −1.35–1.84 | | | | |
| | Chlortetracycline | Antibiotic fate modeling in lakes | LGMF | −0.88–2.13 | | | | |
| | Enrofloxacin | Antibiotic fate modeling in lakes | LGMF | −0.24 −2.54 | | | | |
| | Erythromycin | Antibiotic fate modeling in lakes | LGMF | −1.28 to 2.02 | | | | |
| | Norfloxacin | Antibiotic fate modeling in lakes | LGMF | −1.62–2.53 | | | | |
| | Oxytetracycline | Antibiotic fate modeling in lakes | LGMF | −1.95–1.64 | | | | |
| | Sulfachlorpyridazine | Antibiotic fate modeling in lakes | LGMF | 0.29–1.91 | | | | |
| | Sulfameter | Antibiotic fate modeling in lakes | LGMF | −0.75–2.04 | | | | |
| | Sulfamonomethoxine | Antibiotic fate modeling in lakes | LGMF | −0.24–1.75 | | | | |
| | Sulfamethoxazole | Antibiotic fate modeling in lakes | LGMF | −1.62–1.59 | | | | |
| | Sulfathiazole | Antibiotic fate modeling in lakes | LGMF | −0.92–0.03 | | | | |
| | Tetracycline | Antibiotic fate modeling in lakes | LGMF | −0.22–1.92 | | | | |
| | Trimethoprim | Antibiotic fate modeling in lakes | LGMF | 0.12–2.27 | | | | |
| ePiE | 30 pharmaceuticals | Model validation | MSE | 127 [a] | MSE | ≤150 | UK | [95] |
| | Ibuprofen | Evaluation of model prediction | MSE | 126 | | | UK | [113] |
| | Ibuprofen | Evaluation of model prediction | MSE | 88 | | | German | |
| | Ibuprofen | Evaluation of model prediction | MSE | 866 | | | Spain | |
| | Ibuprofen | Evaluation of model prediction | MSE | 419 | | | Slovenia | |
| | Ibuprofen | Evaluation of model prediction | MSE | 4427 | | | Croatia | |
| EUSES | Dichloromethane | Predict regional concentration | MF | 6 | $R^2$ | ≤0.7 | Japan | [133] |
| | 1,2-dichloroethane | Predict regional concentration | MF | 0.33 | | | | |
| | Triclosan | Predict regional concentration | MF | 0.067 | | | Germany | [137] |
| | Triclosan | Predict local concentration | MF | 16 | | | | |

Notes: $R^2$ is the correlation coefficient between the prediction and measured; MF is the Multiplication Factor, the ratio of predicted to measured concentration, MF < 1 is underprediction, and MF > 1 is overproduction; LGMF is the logarithmic value of MF, LGMG −1, and 1 is supposed to be a good prediction of model; RE is a relative error in percent, RE < 1 is overprediction, and RE > 1 is underprediction; RMSE is the Root Mean Squared Error, zero value RMSE shows ideal representation, and minimum value shows a better picture of the observed statistics; PBIAS is the Percent Bias, negative PBIAS is under prediction and positive PBIAS is overestimation [134]; MSE is the Median Symmetric Error, MSE is the median absolute error where the relative error is defined to have the same direction, MSE assigns equal importance to deviations of the same order rather than the same magnitude, in the pair of predicted with measured concentration; (1, 10) ng/L and (100 ng/L, 1 µg/L) the absolute error is 9 and 900 ng/L but receives equal penalty [95]. More information on MSE can be found in [139].

Application of the same model in different regions has been observed to give different efficiency. Evaluation of ibuprofen prediction using ePiE in different Europe countries (UK, Germany, Spain, Slovenia, and Croatia) using the same simulation scenario has been observed to have variable efficiency among countries. Comparing predicted and observed concentrations gives a median symmetric error of 88% in Germany and 4427% in Croatia. On the other hand, applying the GREAT-ER model in the exact location (Spain) but for

different antibiotics produces a range of discrepancies in efficiency, ranging from a factor of 0.003 to 2 [11]. A similar discrepancy has been observed in regional simulation as well; however, it can be noted that the discrepancy is higher in regional simulation as averaging emission scenarios affect the accuracy of the simulation.

Further, this can be attributed to the spatial discretization, which might be inconsistent with the scale of ground physical phenomena. The phATE in USA, phATE, and iSTEAM in Canada, and GREAT-ER in Spain have been observed to have a relatively lower deviation between the predicted and observed data, as illustrated in Table 2. Further, the deviation in the majority of the models is positive, which is an overestimation. This has been observed mainly in the spatially explicit fate models (phATE, GREAT-ER, ePiE, and EUSES).

Despite the expected acceptable uncertainties, both under and overprediction of the models are gaps that might be linked to various defects and require further investigation and improvement. The description of an area to be modeled requires defining its boundaries, describing the existing conditions and relationship, and characterizing the impact and relationship of the surrounding environment outside the model area. In addition, the characterization of the modeling area must be described in terms of space and time, as well as the selection and characterization of the processes that occur in the system. This necessitates the establishment of relationships and calibration to ensure that the model's perdition is consistent with the monitoring data.

However, as the calibration is dependent on the monitoring data, uncertainties from sampling errors and measurement errors arising from the analytical detection equipment would also affect the prediction of the model. Moreover, due to uncertainties in representing the actual spatial mapping, temporal distribution, boundary conditions, and numerical approximation, the calibration of a model may be burdened with error and uncertainty. In explicit spatial models, due to coarse spatial resolution (e.g., phATE and WASP), model parameterization and site-specific predictions are challenging [140]. Further, site-specific contaminant fate mechanism evaluation is complex. Thus, leading to a range of errors and uncertainty regarding an individual perspective and the application of a model to a different environment. As a result, sensitivity analysis (examination of the impact of model errors and uncertainties) and verification is highly required. However, the spatial and temporal discretization, sensitivity analysis of environmental parameters, emphasis on the spatial distribution of emissions, and the depiction of appropriate spatial patterns of environmental drivers are essential factors that need to be addressed in future research and model development.

The other factor is the conclusive adaptation of a first-order rate constant for all instream degradation mechanisms in natural water bodies. Despite the fact that the second-order decay rate requires extensive mathematical computation, the degradation (e.g., photodegradation, hydrolysis, and sorption) of various antibiotics in various environmental matrices is well expressed through the second-order decaying mechanism [51,58,65,71,141]. As discussed in Section 3, the rate constant of various removal processes is not independent of the particular combination of hydrological and geomorphologic properties of the system under study, which suggests a thorough evaluation of degradation kinetics. Therefore, a particular investigation of the environmental degradation kinetics of given antibiotics in a target area is important.

Accounting variabilities and different scenarios allow for flexible characterization and estimation of different water quality state variables and parameters. Hence, considering variables, such as population, informal settlement, seasonal consumption rate, flow variability, chemical properties, degradation kinetics, sewer leakage, and WWTP removal rates, could help to observe the real phenomenon and improve model prediction [113].

The uncertainty in quantifying antibiotic emissions and physicochemical behavior in the environment makes realistic simulations challenging to obtain. In most cases, conventionally, antibiotic compounds are assumed to be emitted in a standard environment with pre-defined environmental characteristics and use a constant per-capita discharge rate and dilution factors based on country-wide averages. Furthermore, regional-scale level

simulation necessitates a large amount of spatial pattern data over time, which is frequently solved by assuming an even distribution of emission patterns [17]. However, evenly distributed emission patterns do not hold true for local analysis, resulting in significant noise and uncertainty.

The development of models for antibiotics simulation in surface waters is challenged notably by inaccurate boundary conditions, numerical approximation errors, and an inadequate representation of the dynamic physicochemical interaction with various natural and synthetic elements of the aquatic environment. Therefore, future investigations on improving these gaps would greatly help monitor and safely manage the aquatic environment.

## 6. Conclusions

This study reviews antibiotics as an emerging contaminant in the aquatic environment and the current state of the science of environmental modeling. Also reviewed are the sources, occurrence, fate, transport and degradation of antibiotics in the aquatic environment. The natural attenuation mechanisms of antibiotics, physical, chemical, and biological processes, and their complexation with the organic and inorganic matter are presented with a perspective of mathematical modeling of antibiotics. Eleven antibiotic fate models have been evaluated, which are used to simulate contaminant fate from emission to transport and to final sink. To help selection of an appropriate model, a concise review of the mathematical formulations, pollutant simulation, environmental applicability, risk assessment, availability of graphical user interfaces, strength, and limitations of frequently used environmental antibiotic fate models is presented. Additionally, the representative summary of the performances of different model predictions in various countries worldwide has been discussed.

The efficiency of the reviewed models to represent the actual observation of the environmental phenomenon is reviewed upon their corroboration in previous studies. The performance statistics of the models from the validation of a model prediction were collected from various previous studies and discussed to observe their capability and suitability. A respective model's performance depends on the purposes and geographic scale of the simulation. The routine and preliminary evaluation of antibiotic contamination in a river at a catchment and regional level may use a steady-state advection–dispersion model. However, large-scale and exhaustive assessment requires comprehensive research and investigations to predict an observed phenomenon accurately. The spatial and temporal representation, the spatial distribution of emissions, the depiction of appropriate spatial patterns of environmental drivers, and the evaluation of specific removal mechanisms are important system variables to simulate the actual natural occurrence. Apart from uncertainties originating from sampling and measurement errors arising from the analytical detection equipment which would affect the prediction of the model, the sources of discrepancies in most model predictions are also attributed to the inability to capture the effect of variabilities due to flood events, natural degradation kinetics, in-sewer removal, temporal variability of flow, unrealistic calculation of antibiotics emission and WWTP bypass flow. Other than the variable factors, the development of models for antibiotic simulation in surface water is challenged by the inability to represent concentration gradients in poor mixing conditions, inaccurate boundary conditions, error due to numerical approximation, and unrealistic representation of the dynamic physicochemical interaction with various natural and synthetic elements of the aquatic environment. These identified gaps inherent in the existing model call for further rigorous research to monitor and ensure the safety of the aquatic environment.

**Author Contributions:** Conceptualization, T.Z.A., J.T.A., M.K. and M.D.; writing—original draft preparation, T.Z.A.; writing—review and editing, T.Z.A., J.T.A., M.K. and M.D.; supervision, J.T.A., M.K. and M.D.; funding acquisition, J.T.A. All authors have read and agreed to the published version of the manuscript.

**Funding:** This research was funded by Water Research Commission of South Africa, grant number WRC2020-00622. The authors express their gratitude for the support provided by the Water Research Commission in South Africa for this research.

**Data Availability Statement:** No new data were created in this study. Data sharing is not applicable to this article.

**Conflicts of Interest:** The authors declare no conflict of interest.

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
