# Peer review of "Assessment of Existing Fate and Transport Models for Predicting Antibiotic Degradation and Transport in the Aquatic Environment: A Review"

_water, doi:10.3390/w15081511_

Round 1

Reviewer 1 Report

There is very useful information in this review paper.  The units for concentration for equations 1 to 5 should be weight/volume.  The reference 64 listed has concentration in ng/L.   

I would recommend that the information for the models in Tables 1-3 be condensed so the most important information is presented, if possible.  Also, it may be helpful if information about which model evaluation criteria works best with each model and the range of goodness of fit for the different types of models be included, for example, Moriasi has ranges in one of his publications (MODEL EVALUATION GUIDELINES FOR SYSTEMATIC
QUANTIFICATION OF ACCURACY IN WATERSHED SIMULATIONS
D. N. Moriasi, J. G. Arnold, M. W. Van Liew, R. L. Bingner, R. D. Harmel, T. L. Veith).  Criteria for transport models for antibiotic are probably different and it would be helpful to have this information.

Reviewer 2 Report

This review needs a thorough language polish. A major revision is recommended for this manuscript.

Line 20-21 check this sentence

Line 35-36 check this sentence

Line 52-55 rephrase it

Line 164 rephrase it

Line 167 dilution, adsorption, and sedimentation are not in terms of degradation

Line 172 are and is      ?

Line 208 this sentence is nonsense

Line 338-340 rephrase it

Line 376-377 rephrase it

Line 468 change the range of pHto pH

Line 531-533 rephrase it

Line 637-639 rephrase it

Line 655-656 full name should be added at the first appearence of the abbreviations

Table 1 some casing issues

Line 918-919 rephrase it

Table 3 some casing issues

Line 1039-1040 rephrase it

Round 2

Reviewer 2 Report

This manuscript can be accepted now.